# The Authentication and Grading of Edible Bird’s Nest by Metabolite, Nutritional, and Mineral Profiling

**DOI:** 10.3390/foods10071574

**Published:** 2021-07-07

**Authors:** Ramlah Mohamad Ibrahim, Nurul Nadiah Mohamad Nasir, Md Zuki Abu Bakar, Rozi Mahmud, Nor Asma Ab Razak

**Affiliations:** 1Natural Medicine and Products Research Laboratory, Institute of Bioscience, Universiti Putra Malaysia, Serdang 43400, Malaysia; ramlah86ibrahim@gmail.com (R.M.I.); nurul.nadiah.86@gmail.com (N.N.M.N.); zuki@upm.edu.my (M.Z.A.B.); 2Department of Veterinary Pre-Clinical Science, Faculty of Veterinary Medicine, Universiti Putra Malaysia, Serdang 43400, Malaysia; 3Centre for Diagnostic Nuclear Imaging, Universiti Putra Malaysia, Serdang 43400, Malaysia; rozi@upm.edu.my

**Keywords:** food, authentication, edible bird nest, *Aerodramus fuciphagus*, half cup, stripe-shaped, FTIR, GC-MS, nutritional composition

## Abstract

Edible bird’s nest (EBN) produced by *Aerodramus fuciphagus* has a high demand for nutritional and medicinal application throughout the world. The present study was to evaluate the authentication of a man-made house EBN, which are half cup and stripe-shaped by FTIR. Next, both samples were compared according to their metabolite, nutritional, and mineral composition. The results indicated that the FTIR spectra of both EBN samples were identical and similar to the reference, suggesting the authenticity of the EBN used. The metabolites that contribute to the possible medicinal properties of EBN were found by using GC-MS. The results of the proximate analysis, followed by the standard AOAC method, inferred that both EBN shapes to be rich in crude protein and carbohydrate contents. However, the proximate composition between the half cup and stripe-shaped EBN showed significant differences. Major mineral elements detected were calcium and sodium, and magnesium contents were significantly different between both EBN. Additionally, the half cup and stripe-shaped EBN had a low level of heavy metal content than the maximum regulatory limit as set by the Malaysian Food Act 1983. This study concludes that the nutritional composition varied between the samples and thus suggests that nutrient content should be considered as criteria for the grading requirement of commercialized EBN.

## 1. Introduction

Edible bird’s nest (EBN) is traditionally used as one of the best nourishing and comprehensive health food products. It is produced by several different swiftlet species of *Aerodramus fuciphagus* and *Aerodramus maximus* [1]. The nests are constructed from the saliva of male swiftlets, which is secreted from a pair by the sublingual salivary gland during the nesting and breeding seasons. Consumption of EBN can be traced back for hundreds of years and up until now, this tonic food is popular among the Chinese community worldwide. In traditional Chinese medicine, EBN is used as a tonic food to enhance complexion and metabolism, alleviate asthma, treatment of malnutrition, and strengthen the immune system [2,3].

The nutritional values and therapeutic properties of EBN have also been proven through modern science and technology including anti-aging by ameliorating UVB-induced skin photo aging, anti-inflammatory properties such as improving NF-α/IFN-γ-stimulated inflammation and promoting wound healing [4]. EBN is also reported to protect joint degeneration, chondro-protection against osteoarthritis, and enhanced bone strength [5]. It has anti-viral efficacy against influenza viral infection and regulates immune function of the body by promoting lymphocyte transformation [6]. A recent review systemically summarized the neuro-protective activity of EBN in modulating the cognitive performance and functions [7]. EBN is reported to have an anti-oxidant effect against oxidative stress and enhancement of antioxidant capacity [8,9]. Furthermore, EBN has potential benefits on metabolic disorders such as prevent insulin resistance, attenuate diet-induced hypercholesterolemia and improve overall hypertensive effect [10,11] and finally, other therapeutic properties [12]. These pharmacological and nutritional benefits of EBN have been associated with its physicochemical properties.

According to the Malaysian Standard of EBN MS2334; 2011 [13], the classification of EBN into Grades I, II, and III are based on physical properties such as shape, size, degree of cleanliness, and impurities. The grading system standardizes the quality measurement and determines the price of EBN, the higher the grade, the more expensive the EBN. However, some unethical suppliers are reluctant to abide by the standard operating procedures and mix the original EBN with less expensive materials like *Tremella fungus*, agar, fried pigskin, and egg white to increase the size of the nest and market value [14]. The authentication and purity of EBN by Fourier transform infrared spectroscopy (FTIR) involves simple, rapid, and non-destructive methods, requiring no sample preparation to provide molecular information based on the absorption spectrum of various functional groups, reflecting the presence of chemical components by wavelength and intensity.

Edible bird’s nest has a promising medicinal potential due to its valuable biochemical components. The common targets of study interest are the quality and functional properties of EBN, which are often based on protein and carbohydrate contents consisting of approximately 60% and 30% of the total mass, respectively [15]. Apart from these, EBN also contains lipids and other small molecules [16]. However, other possible chemical compounds in EBN are limited. Thus, GC-MS analysis of metabolites is important to contribute a scientific reference for further exploration of the medicinal properties of EBN. Furthermore, identification of metabolites in EBN will provide new insights into understanding the nature of the active principles of EBN and the future discovery of novel bioactive compounds that have therapeutic activities to specific diseases.

A few studies have reported on the nutritional composition of EBN, which were based on various factors such as grade (clean process, and impurities), origin (man-made house or caves), place (Malaysia region or neighboring countries), and color (white, yellow, or red) [17]. The EBN colors following the location of EBN harvested have proven to have some significant differences in its nutritional composition, specifically in its proximate and mineral content. Nevertheless, scientific research on the chemical and nutritional composition of EBN of various shapes and grades is still in paucity and the underexplored area needs further investigation. Therefore, this present study aimed to evaluate the purity, metabolite constituents, and nutrient composition of the two different shapes of man-made house EBN, which is half cup and stripe-shaped.

## 2. Materials and Methods

### 2.1. Samples

The raw, cleaned, and different shapes of EBN samples produced by swiflet species *Aerodramus fuciphagus* were collected in a complete randomized design from three man–made bird houses in a single bird farm located in Terengganu, Malaysia. The sample used was from a mixed portion of several bird nests collected from the three bird houses. In this work, we only studied two different shapes of EBN, which were half cup and stripe-shaped. The half cup EBN was considered the complete shape and it was collected from the edge of the top of a bird house wall. The stripe-shaped EBN was the crushed pieces from the complete EBN and exhibited a hardened texture. They were subsequently cleaned from any feathers or impurities identified during the sample screening procedure and were kept at 4 °C in an air-tight container until further use.

### 2.2. ATR-FTIR Analysis

The authentication and identification of adulterants in EBN were determined by Attenuated total reflectance-Fourier transform infrared spectrometry (ATR-FTIR, Perkin-Elmer Model Spectrum 100, Waltham, MA, USA). One gram of half cup and stripe-shaped EBN sample was subjected to FTIR under a spectral scanning range of 4000~650 cm^−1^ using a miracle ATR technique. The resulting spectrum was used to interpret the characteristics of the functional groups.

### 2.3. Water Extraction

The EBN extracts were prepared based on a previous study [18] with some modification. The EBN sample was ground and soaked in deionized water with ratio of 1:100 (*w*/*v*) and incubated for 16 h at 4 °C. Then, the soaked EBN was double boiled at 100 °C for 30 min following the traditional technique of cooking EBN to preserve the taste, and later cooled to room temperature. It was then filtered through a muslin cloth and the filtrate (EBN water extract) was subsequently frozen in −80 °C prior to freeze drying. The powder form of freeze dried EBN extract was stored at−20 °C until required for further use.

### 2.4. GC-MS Analysis

Water extract of EBN samples was subjected to analysis with GC-MS using the model instrument, GCMS-QP2010 Plus (Shimadzu, Kyoto, Japan), attached with a high polar fused silica capillary column Zebron ZB-FFAP (30 m length × 0.2525 mm I.D × 0.25 μm film thickness). Helium was used as the carrier gas at a flow rate of 1 mL/min and the injector split ratio was set to 1:5. An injection volume of 1 μL was used and the solvent cut-off time was 1.5 min. The injector and source temperatures were set at 200 and 220 °C, respectively. The column oven temperature was set at 100 °C (held for 3 min) and raised 20 °C per min to 210 °C (held for 5 min). The mass spectrometer was operated in electron impact (EI) ionization mode at 70 eV. Data acquisition was performed in the full scan mode from mass-to-charge (*m*/*z*) 35 to 300 with a scan time of 0.5 s with a total run time of 13 min. Retention indices (RI) of the compounds were determined by comparing the retention times of a homologous series of normal alkanes and then calculated as described by van Den Dool and Kratz [19]. Identification of metabolites was performed by matching mass spectra and retention indices to data in the literature and the National Institute of Standards and Technology (NIST08) spectral library. Peaks with mass spectra having a similarity index (SI) higher than 70% were assigned with the respective compound names.

### 2.5. Proximate Analysis

Nutritional compositions, which are moisture, ash, crude fat, crude protein, and carbohydrate, were estimated by using an in-house method based on the Association of Official Analytical Chemists (AOAC, 2000) [20,21]. Three replications were conducted for each analysis.

#### 2.5.1. Moisture Content

The moisture content was determined based on the method as described by the AOAC Method 930.15 (2000) [20] using the standard hot air oven method by drying 4 g of the sample until a constant weight was obtained. The moisture content was calculated as the percent weight of the dry sample (g) divided by the wet weight of the sample (g).

#### 2.5.2. Ash Content

The AOAC Method 942.05 (2000) [20] was used to determine the ash content. A dry ashing method was used to determine the ash content by using 2 g of sample incinerated in a furnace at 550 °C overnight until a white or light grey ash formed. The crucible was cooled in a desiccator, reweighed, and the ash content was calculated as the percent weight of ash (g) divided by the weight of the sample (g).

#### 2.5.3. Crude Fat

Crude fat of ground EBN was determined using the Soxhlet extraction (Soxtec™, FOSS Analytical, Hilleroed, Denmark) based on AOAC Method 920.39 (2000) [20]. Petroleum ether was continuously volatilized and allowed to pass through 10 g of homogenized samples contained in a thimble to extract the ether-soluble matter, which was done automatically through the extraction unit. The extracted matter was collected and was dried at 103 °C for 30 min. Then, it was cooled in a desiccator and weighed as percent weight of fat after evaporation (g) divided by the weight of the sample (g).

#### 2.5.4. Crude Protein

Crude protein of ground EBN was determined using a Kjeltec 2100 instrument (FOSS Analytical, Hilleroed, Denmark) according to the AOAC Method 954.01 (2000) [20]. About 1 g of sample was weighed into digestion tubes containing two copper Kjeltabs (equivalent to 3.5 g K_2_SO_4_± 0.4 g CuSO_4_·5H_2_O) and 12 mL concentrated sulfuric acid (H_2_SO_4_) was added. An exhaust system was attached to the digestion tubes in a rack and a water aspirator was set to get a full effect. The rack was loaded into a preheated digestion block (420 °C), and after 5 min, the water aspirator was turned down until the acid fumes were contained within the exhaust head. Samples were digested until a blue/green solution appeared. The rack of tubes was removed with the exhaust still in place and was left to cool for 15 min. Then, 80 mL of deionized water was added to the tubes. A conical flask containing 25 mL of receiver solution (4% boric acid with bromocresol green and methyl red indicator solution in 95% ethanol) was placed into the distillation unit together with the digestion tubes containing each sample. Sodium hydroxide (40%, 50 mL) was dispensed into the digestion tube and the system valve was opened for approximately 4 min to perform the distillation process. At the end of the distillation cycle, the receiver solution in the conical flask had turned to a green solution, indicating the presence of alkali (ammonia). The distillate in the conical flask was titrated with hydrochloric acid (HCl, 0.1 N) until the blue/grey endpoint was achieved. The volume of HCl consumed in the titration was recorded. The same procedure was repeated for the blank. The percent nitrogen was calculated by different volume titration over blank multiply 0.1 N HCI multiply 14.007 divided by the weight of the sample used (g). Next, the crude protein of the samples was calculated by percent nitrogen to multiply the crude protein factor of 6.25.

#### 2.5.5. Carbohydrate Content

The carbohydrate content of ground EBN was determined by the difference method described by Idris et al. (2019) [21]. The percentage of total carbohydrates was determined by reducing the sum percentages of moisture, ash, crude fat, and crude protein to one hundred.

#### 2.5.6. Total Dietary Fiber

The total dietary fiber content of the ground EBN sample was determined by a Fibertec extraction system (FOSS Analytical, Hilleroed, Denmark) based on the AOAC Method 991.43 (2000) [20]. One gram of sample was added with phosphate buffer solution (50 mL, pH 6.0), then the pH 6.0 was adjusted by adding 0.275 N sodium hydroxide (NaOH) or 0.325 N HCl. Alpha-amylase (50 µL) was added and incubated at 100 °C in the water bath for 30 min. The sample was cooled at room temperature and the pH was adjusted to 7.5 with 0.275 N NaOH, after which 100 µL protease was added to the sample and further incubated in the water bath with continuous agitation at 60 °C for 30 min. The sample was cooled at room temperature, and again, their pH was adjusted to pH 4.0–4.6 by adding 200 µL amyloglucosidase. The sample was continuously stirred and incubated in a water bath with continuous agitation at 60 °C for 30 min. Then, the enzyme-digested sample was filtered through the crucible containing a celite^®^ 545 bed into the receiver flask. A crucible was dried overnight at 105 °C and cooled in a desiccator for an hour. The crucible containing dietary fiber was weighed and the residue weight was calculated as 100 timesthe weight of the mean residue (g) minus protein residues (g) minus ash residues (g).

#### 2.5.7. Caloric Value

The caloric value was calculated by the sum of the percentages of protein and carbohydrate multiplied by a factor of 4 (kcal/g), total dietary fiber multiply by a factor of 2 (kcal/g), and crude fat multiply by a factor of 9 (kcal/g) [21].

### 2.6. Major and Trace Elemental Analysis

Major and trace minerals in EBN samples were analyzed using inductively coupled plasma-optical emission spectroscopy (ICP-OES) based on AOAC method 985.35 (2000) [20]. Fifteen minerals and trace elements, namely, boron, calcium, potassium, magnesium, sodium, phosphorus, sulfur, iron, manganese, copper, chromium, selenium, molybdenum, and zinc were analyzed. About 1 g of EBN ground sample was digested in a mixture of 65% nitric acid (HNO_3_) and 30% hydrogen peroxide (H_2_O_2_) with a microwave digester (CEM Corporation, Matthews, NC, USA). The digestion was carried out at 220 °C for 45 min until a clear transparent solution was obtained. The digest was then made up to 50 mL with a 2% HNO_3_ solution and analyzed with ICP-OES (Perkin Elmer optima 7000DV) equipped with an S10 auto-sampler and WinLab32TM for ICP V5.1 (Perkin Elmer, Waltham, MA, USA). The calibration was performed with a standard mixture from Perkin Elmer (Waltham, MA, USA) and all elements were determined by the axial plasma view. The analytical method used to determine elemental content in the samples was well established and accredited under the Laboratory Accreditation Scheme of Malaysia (SAMM), which is in accordance with recognized International Standard ISO/IEC 17025. For each sample analysis, three replicates were measured in order to assure the control quality of our measurement.

### 2.7. Heavy Metal Analysis

Inductively coupled-plasma mass spectrometry (ICP-MS) based on AOAC method 9.1.09 (2000) [20] was used to analyze the heavy metal content in EBN samples. The heavy metals analyzed were arsenic, cadmium, mercury, nickel, and lead. Five hundred milligram EBN samples were digested with a microwave digester coupled with an SK-10 high-pressure rotor and analyzed on the ICP-MS (Perkin Elmer, Waltham, MA, USA) [10]. The ICP-MS system was calibrated using a blank consisting of multi-element standards in 1% nitric acid at concentrations ranging from 5–200 µg/L and 1% nitric acid in ultrapure water. This analytical method was accredited under SAMM and follows ISO/IEC 17205 standards. Three replications were conducted for this analysis.

### 2.8. Statistical Analysis

The results were reported as mean ± standard deviation. The statistical comparison among the groups was performed with the Student *t*-test using SPSS 20.0 software (SPSS Inc., Chicago, IL, USA). Differences were considered significant when the *p*-value < 0.05.

## 3. Results and Discussion

### 3.1. FTIR Analysis

The ATR-FTIR spectra of the half cup and stripe-shaped EBN can be observed with absorption bands in the range of 4000–650 cm^−1^, as presented in Figure 1. The spectra patterns of both EBN shapes were similar, indicating the presence of similar functional groups in the EBN samples.

The sample spectra pattern of this present finding was comparable with the reference spectra pattern previously studied by Hamzah et al. (2013) [22], as shown in Table 1. The presence of hydroxyl, alkanes, alkynes, amides, aldehydes, amine, and carboxylic functional groups was confirmed in both studies, except in the fingerprint regions of the spectra at 1740–1720 cm^−1^ and 895–885 cm^−1^. Differences found in the absorption band from these studies imply that the EBN spectrum changes according to factors such as source, habitat, or processing conditions [23].

A genuine EBN is largely constituted of glycoprotein, which can be observed in the region of 1695–1630 cm^−1^ with C=O stretching of amide I and 1560–1500 cm^−1^ with N–H bending of amide II attributed to protein. The absorption band peaks of the protein were observed to be similar in both shapes. This finding is similar to another report, which showed that the most characteristic FTIR absorption bands of pure EBN were found at about 1640–1320 cm^−1^, representing a protein, which is the amide, mono-substitution amide, and di-substitution amide [24]. The absorption band in the region of 1250–1020 cm^−1^ due to C–N stretching of amine and 1440–1395 cm^−1^ of carboxylic acid are functional groups of amino acid content in EBN. Other absorption bands at the region of 1390–1380 cm^−1^ and 1085–1050 cm^−1^ in half cup and stripe-shaped EBN due to the vibration of aldehyde and hydroxyl groups, indicates the presence of carbohydrate. While the broad FTIR spectrum in the region of 3550–3200 cm^−1^, representing the O–H stretching functional group of water, which present in both shapes of EBN indicate the presence of moisture that may be residual from the washing and soaking process upon cleaning the EBN. There were signals detected in the region of 895–885 cm^−1^, similar to a recent finding by Gan et al. (2020) [25], which were due to the out-of-plane bending of aromatic C–H bonds. The authenticity of a pure EBN is mainly reflected in the absorption peaks of its major compound. In this study, both the EBN samples possessed protein and carbohydrate peaks indicating their authenticity.

Fourier transform infrared spectroscopy quantifies the absorption of infrared light by molecule, resulting in a specific FTIR spectrum that reflects the overall composition of the sample by wavelength and intensity. The IR spectroscopy measures the chemical covalent bonds and thus creates a molecular fingerprint of the chemicals present in a sample [26]. The use of FTIR spectroscopy in food analysis has become more attractive because of its cost-effective nature, non-destructive measurements as well as convenience for screening purposes. This analysis has been established to be useful for adulteration detection and quantification in food products [27].

### 3.2. GC-MS Analysis

The analysis of EBN with GC-MS revealed that five metabolites were identified in both half-cup and stripe-shaped EBN water extracts, as shown in Figure 2a,b. EBN water extract was used in this study because EBN contains mostly water soluble compounds and aqueous extraction is the most commonly used method in EBN extraction [28].

Table 2 presents the identified compounds with their retention time (min), retention indices (RI), similarity index (%), and peak area (%). It was revealed that the most abundant metabolite in the extract of half cup EBN was cyclobutanol and for stripe-shaped EBN, it was 1-propanol, 2-amino. The other metabolites were similar for both EBN shapes in the GC-MS analysis. The identified metabolites in this analysis belong to different chemical classes such as cycloalkane (cyclobutanol), amino alcohol (1-propanol, 2-amino), carboxylic acid (acetic acid, formic acid, ethyl 2-hydroxypropanoate), and amide (acetamide).

Literature data on the metabolite profile in EBN is very limited, thus the present study was non-comparable to any of the reported data. It is important to identify the metabolite composition in EBN because it opens up the possibility of new explanations for the health benefits of EBN. A few studies have reported the anti-bacterial properties of cyclobutanol, which may support the findings of similar activity in EBN [29,30]. Formic acid and acetic acid are systematically the simplest carboxylic acids and are reported to have an antibacterial effect against *Escherichia coli* and *Staphylococcus aureus* [31] and microbial inhibitory effects at different levels of contamination by *Salmonella typhimurium* [32], respectively. Hence, the identification of this metabolite may support the finding on the antibacterial effects of EBN against food borne pathogens [33]. Acetamide is a simple amide with the chemical formula C_2_H_5_NO, which may be formed naturally in food or by the product of other processes. The presence of acetamide in the EBN samples may be due to the abundance of N-acetylated moieties on sugar, amino acid, and protein, which is N-acetylneuraminic, also known as sialic acid, which shows the characteristics of EBN [34]. Ethyl 2-hydroxypropanoate is a low molecular weight organic acid found naturally in a wide variety of foods and exhibits features such as skin hydration, skin rejuvenating, skin lightening, pH regulators, and anti-acne agent. This may probably explain the skin lightening activities and anti-aging effect of EBN, together with other bioactive components in the extracts [4]. The functional properties of 1-propanol, 2-amino are unknown and need further experimental investigation to exploit the functional potential of this compound. The identification of these metabolites provided logical explanations to some of the claimed medicinal effects of EBN as mentioned earlier, however, in-depth studies need to be performed to further explore each metabolite’s properties on its reputable medicinal benefits.

### 3.3. Proximate Analysis

The proximate composition of different shaped EBN, which are half cup and stripe-shaped, is shown in Table 3. This present finding of proximate composition was in the range of the other reported literature [15,35], with protein being the major content, followed by carbohydrate, moisture, and ash, while no fat was detected (lower than the limit of detection, LOD). Interestingly, fiber was found to be present in both EBN. The proximate composition between the different shapes of EBN was found to be significantly different (*p* < 0.05).

It was observed that half cup EBN had a higher content of protein (56.96 ± 0.09%) than stripe-shaped (54.70 ± 0.16%), with a significant difference. However, the range of protein content for both EBN showed quite similar data to the previous study that stated that the protein content range was between 56.20–61.50% [36]. The carbohydrate content of the half cup EBN was significantly different from stripe-shaped EBN with 23.96 ± 0.13 and 22.12 ± 0.18%, respectively. The carbohydrate content in the EBN was found to be similar to previous reports that ranged between 10.63–31.40% [15]. The higher protein and carbohydrate content in half cup EBN can be possibly explained due to the complete structure of the nest, which was composed almost entirely of pure mucin-rich glycoprotein that hardens in contact with air, forming a cup-shaped nest with little impurities [37]. In contrast, stripe-shaped EBN are the pieces collected from the edge of the EBN nest, which adheres to the surface of the bird house and have more impurities trapped to its harden texture. Stripe-shaped EBN required a tedious cleaning process and caused some nutritional loss throughout the process. Analysis of EBN using chemical methods showed that EBN is a mucin glycoprotein with properties of both protein and carbohydrate [38].

Moisture content found in the half cup and stripe-shaped EBN was 15.92 ± 0.08% and 19.51 ± 0.04%, respectively, which was relatively high compared with other studies, ranging from 7.50–14.00% [15,35]. However, another study had a similar range of moisture content as reported in this study (17.8–24.3%) [39]. The moisture level was likely affected by the cleaning and drying process of EBN since the hardened texture of the stripe-shaped EBN required a longer soaking time in the water to soften them for the removal of feathers and plumages. The ash content of half cup EBN (3.16 ± 0.04%) was significantly lower than stripe-shaped EBN (3.67 ± 0.03%) and it was within the range as reported by previous studies (2.1–7.4%) for house EBN [15,39]. The higher ash content in stripe-shaped EBN could be due to the presence of feathers and other foreign matter trapped in the hardened part of the nest and the loss of water-soluble minerals during the cleaning process [40].

There was a significant difference in fiber content between half-cup EBN (3.89 ± 0.80%) and stripe-shaped (19.96 ± 0.38%) and it was in contrast with the findings from the literature, which reported fibers between 0.10–0.70% [41]. The fiber content in stripe-shaped EBN may consist of high fibrous proteins such as keratin, collagen, or vegetative material from nutrients available for the swiftlet in the particular environment [42]. Caloric value is the energy accumulated in food substances due to protein, carbohydrate, fat, and fiber content. There was a significant difference in caloric value between half-cup EBN (331.00 ± 1.41 kcal/100 g) and stripe-shaped (349.50 ± 0.71 kcal/100 g), which was attributed to the higher fiber content in the stripe-shaped EBN.

### 3.4. Major and Trace Elements

The major and trace elements detected in the half cup and stripe-shaped EBN are presented in Table 4 and compared to other studies from the literature [15,43]. Both shapes of EBN showed relatively high amounts of macronutrients, particularly calcium, sodium, sulfur, magnesium, and potassium, followed by other macro and trace elements. For macro-elements, calcium, magnesium and sodium, were found to be significantly different between the half cup and stripe-shaped EBN, whereas iron and zinc content between the samples were significantly different for the trace elements.

The most abundant macro-mineral in half cup EBN is calcium (735.45 ± 4.38 mg/100 g), which was significantly different from stripe-shaped EBN (652.95 ± 6.40 mg/100 g). The highest level of sodium and magnesium with a significant difference was found in the stripe-shaped EBN (682.14 ± 3.86 and 129.57 ± 0.01 mg/100 g) compared to half cup EBN (504.90 ± 0.46 and 105.97 ± 1.50 mg/100 g). The sulfur content was found to be higher in the half cup (240.16 ± 2.55 mg/100 g) than in the stripe-shaped EBN (211.32 ± 17.38 mg/100 g) with no significant difference between the shapes. Other major minerals such as potassium > phosphorus > boron were found to be present in both shapes of EBN with no significant difference. The present finding for the major elements was in agreement with other studies as reported in the literature, except for some values that differed, which may be due to the method and instruments used [15,44].

The order of highest to the lowest level of trace element content such as copper > iron > zinc > manganese > chromium > selenium > molybdenum > cobalt were the same in both the half cup and stripe-shaped EBN. Stripe-shaped EBN has a significantly higher content of copper (2.34 ± 0.17 mg/100g), iron (1.13 ± 0.03 mg/100 g) and zinc (0.76 ± 0.01 mg/100 g) than the half cup EBN (0.88 ± 0.09, 0.68 ± 0.10 and 0.44 ± 0.05 mg/100 g, respectively). Chromium, cobalt, molybdenum, selenium, and manganese were not significantly different between the EBN samples. The trace element content of EBN in our study was within the range of micro-element content of another reported study [43].

As the amount of ash content in the half cup and stripe-shaped EBN was different, some of the mineral content found in both the EBN shapes also differed. This is because ash is the inorganic residue remaining after a sample is completely burned, which provides a measure of the total amount of minerals within a food [41]. Some of the possible suggestions on the difference of the mineral content between a half cup and stripe-shaped EBN are either that they are naturally produced by the swiftlet themselves, the availability of food source for the swiftlet, or leached from the environment [45]. Overall, all the elemental content in different shapes of EBN were within the range of recommended nutrient intake [46], suggesting EBN as a good source of multi-minerals.

### 3.5. Heavy Metal Content

The heavy metal content in the half cup and stripe-shaped EBN is presented in Table 5 and compared to the maximum permissible proportion of heavy metal in specified food (EBN) set by the Food Act 1983 [47,48] and provisional tolerable weekly intake (PTWI) set by the Food and Agriculture Organization/World Health Organization (FAO/WHO) Joint Expert Committee on Food Additives (JECFA) [49]. The results showed that the lead level was not significantly different between the half cup (116.61 ± 34.98 ppb) and stripe-shaped (115.21 ± 29.25 ppb) EBN and it was below the permitted level of 300.00 ppb. Meanwhile, the arsenic level in half cup EBN (8.76 ± 1.46 ppb) and stripe-shaped EBN (23.81 ± 2.15 ppb) was significantly different between the shapes, but were still lower than the permitted level for EBN (150.00 ppb). The trace amount of cadmium level, 3.15 ± 2.99 ppb, was found only in the EBN half cup and not detected in the stripe-shaped EBN. The range of heavy metal concentration of this finding was in accordance with Chen et al. (2014) [44] for lead (2.24–592.84 ppb), arsenic (0.06–34.35 ppb), and cadmium (0.06–1.87 ppb). Other heavy metals such as mercury and nickel were not detected in both of the EBN samples.

To evaluate potential hazards resulting from long-term daily consumption of EBN containing these measured heavy metals, the concept of an estimated daily intake (EDI) was calculated and used as a reference. As there were no reported data about the EBN daily consumption among consumers, we estimated that people consumed about one piece of half cup EBN daily, which roughly weighs around 5–6 g. Based on this amount, the estimated weekly consumption of our EBN for an adult was still within the PTWI and considered safe for consumption among consumers. The PTWI by the FAO/WHO for arsenic, cadmium, lead, mercury, and nickel were 0.015, 0.007, 0.025, 0.0016, and 0.035 mg/kg body weight [49].

Heavy metal contamination in EBN could be from within the swiftlet house such as rusty iron bars, pressure-treated wood, and lead-based paints that were used in the construction of the bird house or during the processing and manufacturing processes of EBN involving chemicals and materials that could be the source of heavy metals [50]. Other possible contamination may be due to various depositions of heavy metals in the feathers of the swiftlet during their exposure to the external environment and the impurities remained in the nest despite the cleaning process [44]. This finding suggests that the half cup and stripe-shaped EBN concentration is less than the permitted level for the metals studied and is safe for human consumption. However, we still recommend a moderate consumption of EBN in order to achieve an optimal nutrient intake and to lower possible health risks due to the concern of the long-term cumulative effects of these heavy metals, which can cause various diseases and disorders [51].

## 4. Conclusions

In conclusion, the present study showed that the rapid determination of EBN purity by using FTIR-ATR spectroscopy was identical between the different shapes of EBN, suggesting that similar functional groups of genuine EBN were present in both samples. The characteristic of pure EBN in both shapes was confirmed with the presence of amide peaks attributed to protein, hydroxyl, and aldehyde peaks denoted to carbohydrates. The approach of metabolite identification using GC-MS analysis provides new insights on the EBN properties and logical explanations to some of the claimed medicinal effects of EBN. This finding also showed that half cup EBN has significant sources of protein, carbohydrate, and other nutrients than stripe-shaped EBN. Furthermore, different shapes of EBN possess significantly different nutritional compositions in terms of proximate content and some of the minerals attributed to the nature of the EBN itself, which are half-cup and stripe-shaped, the texture of the EBN (soft, hard), and the impurities present. Additionally, both EBN shapes are also safe for consumption since the heavy metals found in this study were below the permitted level. Finally, it is recommended that nutritional composition should be used as one of the grading criteria of EBN, realizing the inconsistency that occurs in human judgment.

## Figures and Tables

**Figure 1 foods-10-01574-f001:**
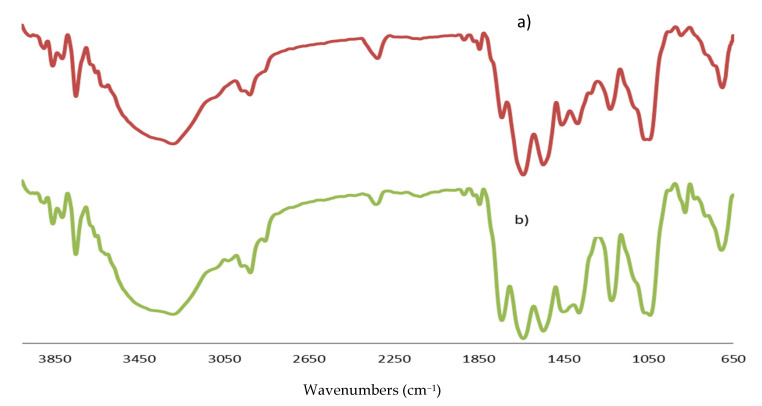
FTIR spectra of EBN (**a**) half cup and (**b**) stripe-shaped.

**Figure 2 foods-10-01574-f002:**
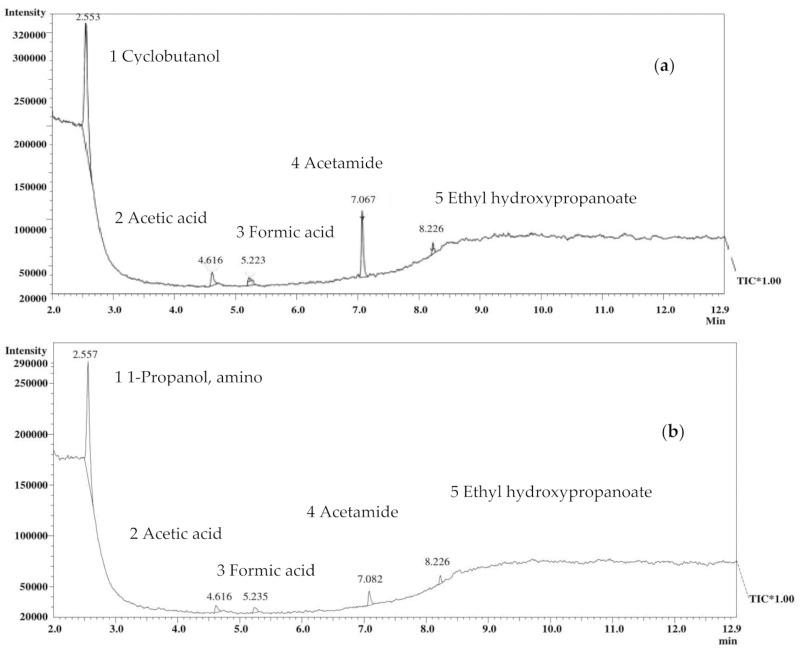
Total ion chromatogram of the water extract EBN of (**a**) half cup and (**b**) stripe-shaped analyzed with GC-MS.

**Table 1 foods-10-01574-t001:** Absorption range (cm^−1^) and functional groups of the half cup and stripe-shaped EBN by FTIR.

Characteristic Absorptions (cm^−1^)	Functional Groups	Compound Class	Absorption Ranges (cm^−1^)		
			Present Study		[22]
			Half Cup EBN	Stripe-Shaped EBN	Processed EBN
3550–3200	O-H stretch	Alcohol	3288.55	3288.48	3381.70
3000–2840	C-H stretch	Alkane	2927.02	2924.37	2931.40
2140–2100	C≡C stretch	Alkyne	2125.60	2129.68	2131.10
1740–1720	C=O stretch	Aldehyde	1738.74	1739.50	ND
1695–1630	C=O stretch	Amide I	1638.59	1638.38	1643.30
1560–1500	N-H bend	Amide II	1544.17	1544.09	1544.50
1440–1395	O-H bend	Carboxylic acid	1406.56	1401.08	1443.90
1390–1380	C-H bend	Aldehyde	1382.82	1376.19	1399.60
1250–1020	C-N stretch	Amine	1228.55	1222.20	1317.80
1085–1050	C-O stretch	Primary alcohol	1054.80	1051.10	1034.80
895–885	C=C bend	Alkene	895.29	886.06	ND

ND: Not detected.

**Table 2 foods-10-01574-t002:** Metabolite composition of the half cup and stripe-shaped EBN analyzed by GC-MS.

EBN	Peak	Retention Time (min)	Retention Indices	Compound Name	Similarity Index (%)	Normalized Area (%)	Compound Nature
Half cup	1	2.553	668	Cyclobutanol	84	61.64	Cycloalkanes
	2	4.616	576	Acetic acid	98	6.66	Carboxylic acid
	3	5.223	933	Formic acid	98	4.87	Carboxylic acid
	4	7.067	629	Acetamide	99	23.68	Amide
	5	8.226	848	Ethyl 2-hydroxypropanoate	86	3.14	Carboxylic acid
Stripe-shaped	1	2.557	741	1-Propanol, 2-amino	91	78.58	Amino alcohol
	2	4.616	576	Acetic acid	95	5.15	Carboxylic acid
	3	5.235	980	Formic acid	96	4.40	Carboxylic acid
	4	7.082	629	Acetamide	97	8.26	Amide
	5	8.226	848	Ethyl 2-hydroxypropanoate	89	3.60	Carboxylic acid

**Table 3 foods-10-01574-t003:** Proximate composition and total dietary fiber content of the half cup and stripe-shaped EBN.

Nutritional Composition (%)	Present Study		[35]	[15]
	EBN Half Cup	EBN Stripe-Shaped	House EBN	House EBN
Crude Protein	56.96 ± 0.09 ^a^	54.70 ± 0.16 ^b^	42.00–63.00	53.00–56.40
Carbohydrate	23.96 ± 0.13 ^a^	22.12 ± 0.18 ^b^	10.63–27.26	28.00–31.70
Moisture	15.92 ± 0.08 ^b^	19.51 ± 0.04 ^a^	7.50–12.90	10.80–14.00
Ash	3.16 ± 0.04 ^b^	3.67 ± 0.03 ^a^	2.10–7.30	2.20–3.40
Fiber	3.89 ± 0.80 ^b^	19.96 ± 0.38 ^a^	NA	NA
Crude Fat	ND	ND	0.14–1.28	0.1
Caloric value	331.00 ± 1.41 ^b^	349.50 ± 0.7 ^a^	NA	NA

Values are expressed as mean (%) ± S.D. Means with different superscript letters in a row indicate a significant difference between a half cup and stripe-shaped EBN (*p* < 0.05). ND: Not detected (lower than Limit of Detection, LOD), NA: Not available.

**Table 4 foods-10-01574-t004:** Major and trace element (TE) content of the half cup and stripe-shaped EBN.

Element Content(mg/100 g)		Present Study		[15,43]	RNI-Tolerable Upper Intake Level (mg/day)[46]
		Half Cup EBN	Stripe-Shaped EBN	House EBN	
Major element	Boron	0.03 ± 0.01 ^a^	0.04 ± 0.01 ^a^	NA	Not Set
	Calcium	735.45 ± 4.38 ^a^	652.95 ± 6.40 ^b^	123.10–859.80	2500.00
	Magnesium	105.97 ± 1.50 ^b^	129.57 ± 0.01 ^a^	88.30–152.80	350.00
	Potassium	16.72 ± 0.22 ^a^	22.00 ± 2.48 ^a^	3.64–35.20	Not Set
	Phosphorus	1.95 ± 0.30 ^a^	4.10 ± 0.74 ^a^	0.03–6.79	4000.00
	Sodium	504.90 ± 0.46 ^b^	682.14 ± 13.86 ^a^	263.80–670.80	2300.00
	Sulfur	240.16 ± 2.55 ^a^	211.32 ± 17.38 ^a^	624.40–884.00	Not Set
Trace element	Chromium	0.03 ± 37.15 ^a^	0.04 ± 28.87 ^a^	0.01–0.06	Not Set
	Cobalt	0.001 ± 2.74 ^a^	0.001 ± 1.91 ^a^	0.00–0.06	0.04
	Copper	0.88 ± 0.09 ^b^	2.34 ± 0.17 ^a^	0.47–11.06	10.00
	Iron	0.68 ± 0.12 ^b^	1.13 ± 0.03 ^a^	0.16–1.94	45.00
	Molybdenum	0.002 ± 0.29 ^a^	0.002 ± 1.56 ^a^	0.00–0.09	2.00
	Selenium	0.01 ± 3.23 ^a^	0.01 ± 0.87 ^a^	0.01–0.04	0.40
	Zinc	0.44 ± 0.05 ^b^	0.76 ± 0.01 ^a^	0.05–2.26	45.00
	Manganese	0.14 ± 0.08 ^a^	0.14 ± 0.02 ^a^	0.02–0.59	9.00

Values are expressed as mean (mg/100 g) ± S.D. Means with different superscript letters in a row indicate a significant difference between a half cup and stripe-shaped EBN (*p* < 0.05). NA: Not available.

**Table 5 foods-10-01574-t005:** Heavy metal content of the half cup and stripe-shaped EBN.

Heavy Metal Content (pbb)	Present Study		[44]	Maximum Regulatory Limit
	Half Cup EBN	Stripe-Shaped EBN	House EBN	
Arsenic	8.76 ± 1.46 ^b^	23.81 ± 2.15 ^a^	0.06–34.35	150.00 *
Cadmium	3.15 ± 2.99	ND	0.06–1.87	1000.00 **
Lead	116.61 ± 34.98 ^a^	115.21 ± 29.25 ^a^	2.24–592.84	300.00 *
Mercury	ND	ND	0.06–70.18	70.00 *
Nickel	ND	ND	56.14–400.00	500.00 ***

All values are expressed as mean (ppb) ± S.D. * Values are permissible levels set [48]. ** Values are permissible levels set [47]. *** Values obtained [44]. Means with different superscript letters in a row indicate a significant difference between a half cup and stripe-shaped EBN (*p* < 0.05). ND: Not detected (lower than LOD).

## Data Availability

Not applicable.

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
