# Peer review of "The Authentication and Grading of Edible Bird’s Nest by Metabolite, Nutritional, and Mineral Profiling"

_foods, 2021, doi:10.3390/foods10071574_

Round 1
Reviewer 1 Report
The comment are present in the attached file. Thanks

Author Response
Please see the response in the attachment.

Reviewer 2 Report
Reviewer's comment on Manuscript Number: foods-1204899
The manuscript aim was to evaluate the purity, metabolite constituents, and nutrient composition between the different shapes of house edible bird’s nest (EBN), which is a half cup and stripe-shaped. The subject of the manuscript falls within the scope of Foods. The results are interesting and valuable especially for Chinese population, which is the main consumer of
I have a few questions concerning the manuscript:
- What are the types and shapes of edible bird’s nest? The introduction lacks this information.
- How many samples of each type were analyzed?
- Why water extract was used for GC MS, where injection of water is not recommended? How the extraction was performed – the manuscript lacks description
- What was the aim of GC MS analysis? What can be reason for finding such metabolites?
- Were methods of analysis validated? Such protocol should be described. Especially in case of minerals data should be given, including LOD and LOQ as some heavy metals were not detected (in such case LOD should be given). Without validation data comparison with literature data should not be performed.
- What is the consumption of EBN and the consumers’ risk of intoxication in view of PTWI?
Author Response

(The authors gave the same response as above.)

Round 2
Reviewer 2 Report
I propose to accept the manuscript as it is.